# Activating silicon for high hydrogen conversion and sustainable anode recovery

Mili Liu [1,5], Yunqi Jia[1,5], Jiangwen Liu[1], Kang Chen[1], Hao Zhong[1], Lin Jiang [2] ✉, Hui Liu[1,3] ✉, Liuzhang Ouyang[1,4] ✉ & Min Zhu[1] ✉

The hydrolysis/methanolysis of silicon has received considerable attention to achieve efficient and on-demand hydrogen conversion. However, the intense covalent network and highly localized electrons in pure Si impede its reactivity with water ($H_2O$) or methanol ($CH_3OH$), thereby hindering the hydrogen release. In this work, we report the synthesis of Zintl phase alkalis-Si alloys via simple ball-milling or sintering, showing eminent performance in enhancement of $H_2O/CH_3OH$ dissociation. Experiments combined with DFT calculations have revealed that the obtained Zintl phase alloys exhibit discrete Si clusters containing well-defined unpaired electrons that efficiently facilitate the interaction between reductant and solvent molecules. Such an effect thereby reduces the activation barrier of $H_2O/CH_3OH$ dissociation to yield active intermediates containing Si-H structure, which significantly promotes the hydrogen release with favorable kinetics and efficiency. The optimal Zintl $Li_{21}Si_5$ alloy achieves ultrahigh Si utilization rates of 86.9% in water and 98.1% in methanol at 25 °C, respectively. Remarkably, even at an extremely low temperature of −40 °C, a substantial hydrogen yield of 1.091 L g$^{-1}$ in methanol is retained. Furthermore, the desirable Zintl phase-water reaction inspires an economic-friendly "charge-hydrolysis-separation" strategy, for effectively recovering the valuable lithium, graphite, Si and Cu resources from the degraded lithium-ion batteries.

Hydrogen, as a carbon-free energy carrier with high energy density[1,2], emerges as a promising solution for sustainable development[3–5]. The hydrolysis or methanolysis pathways, which are driven by the chemical dissociation of water or methanol molecules (HORs, R presents H or $CH_3$ radical), offer competitive strategies for portable green hydrogen production[6]. These approaches feature controllable and on-demand $H_2$ support without the requirement of complex devices or electricity input[7,8], feasibly realizing on-site $H_2$ source disposal without risky storage or transport[9]. Magnesium[10–12], aluminium[13–16] and calcium metals[17] have been widely-used as the reductants for these methods,

where 1 gram material can provide 0.050 – 0.111 mol free valence electrons to interact with HORs molecules for releasing hydrogen gas. In comparison, silicon (Si) serves as a metalloid with notable advantages due to its four valence electrons. Namely, Si has 0.142 mol free valence electrons per gram of material, supplying more electrons to interact with solvent molecules to release $H_2$[18–20]. Theoretically, the Si-HORs reaction involves a crucial transition step where the solvent molecules actively break the H-O bonds of hydroxide radicals, promoting the formation of RO-Si··-Si-H intermediates[21,22]. Subsequently, these active Si-H structure interacts with additional solvent molecules

[1]School of Materials Science and Engineering, Guangdong Provincial Key Laboratory of Advanced Energy Storage Materials, South China University of Technology, Guangzhou, PR China. [2]School of Microelectronics, Shanghai University, Shanghai, PR China. [3]School of Chemistry and Material Science, Hunan Agricultural University, Changsha, PR China. [4]Guangdong Engineering Technology Research Center of Advanced Energy Storage Materials, Guangzhou, PR China. [5]These authors contributed equally: Mili Liu, Yunqi Jia. ✉e-mail: linjiang@shu.edu.cn; liu.hui@hunau.edu.cn; meouyang@scut.edu.cn; memzhu@scut.edu.cn

to rapidly release hydrogen gas. However, the practical reaction of Si-HORs at room temperature rarely progresses as anticipated[23–26], resulting in minimal hydrogen gas release owing to that each Si atom shares valence electrons with neighboring atoms to form a stable octet in the outermost shell. The strong Si-Si covalent bonds make it challenging to cleave electron pairs, thereby hindering the silicon's ability to donate valence electrons and interact with surrounding solvent molecules.

Various strategies have been explored to enhance $H_2$ production from silicon involved HORs[18,27]. The modification of Si using various complex acids[18,19,28,29] has been commonly employed to remove the native oxide layer and construct Si-H structure on the surface[30]. These modifications seek to address concerns regarding the activation of solvent molecules. However, the reactivity of the inner Si matrix for $H_2$ generation still remains limited once the surface Si-H species are consumed by reacting with solvent molecules[31,32]. Therefore, enhancing the intrinsic reactivity of Si emerges as a more viable strategy compared to merely increasing Si atom utilization via surface modification.

To this end, we have focused on the alteration of the Si atom's electronic structure to activate its reactivity. In specific, alkalis-Si (A-Si) alloys in Zintl phases are of particular interest in this regard. Contrary to the pure Si with a robust covalent bonding network and fixed valence electrons, Zintl Si-based phases possess loose Si clusters with unpaired electrons caused by the electron transfer from alkaline atoms to the Si atoms[33]. Such elaborate designs would be particularly beneficial for $H_2$ production, where the local unpaired electrons on the Si site efficiently support the electron interaction with HORs molecules to break H-O bonds of hydroxide radicals.

Herein, we report the construction of five Zintl Si-based phase alloys, including four Li-Si alloys ($Li_{12}Si_7$, $Li_7Si_3$, $Li_{13}Si_4$ and $Li_{21}Si_5$), and one NaSi alloy. These alloys feature various metal-silicon quasi-ionic connections and serve as model systems for the cleavage of $H_2O$ and $CH_3OH$ to release hydrogen. The optimal Zintl $Li_{21}Si_5$ alloy achieves excellent Si utilization rates of 86.9% in water and 98.1% in methanol at room temperature, whereas no hydrogen release is observed from pure Si. To the best of our knowledge, the methanolysis system utilizing the Zintl $Li_{21}Si_5$ alloy demonstrates extraordinarily high hydrogen generation performance beyond the state-of-the-art technology at an ultra-low operating temperature of −40 °C. Mechanism studies indicate that Zintl phase alloys can significantly reduce the active barrier of $H_2O$/$CH_3OH$ dissociation, leading to the formation of advantageous Si-H intermediates. This reduction is attributed to the presence of hyperactive unpaired electrons within the alkalis-silicon structure. Notably, the Zintl $Li_{21}Si_5$ alloy exhibits ultra-low activation barrier energies of −1.13 and −1.47 eV, which are substantially lower than those of pure Si (1.47 eV and 1.24 eV). Moreover, the high efficiency of Zintl-HORs reaction inspires the green anode materials recovery and hydrogen production from the degraded Si-based lithium-ion batteries (LIBs) through a developed "charge-hydrolysis-separation" technology, which also promises a strategy for guiding scrapped Si as a valuable resource to support a post-circular economy.

## Results

### Design concept and construction of Zintl Si-based phase alloys
To elucidate the advantages of Zintl phases on hydrogen production from HORs dissociation, the structural characteristics and electronic states of Zintl Li-Si/NaSi alloys and pure silicon were elucidated by density functional theory (DFT) calculations. The electron density of states (DOS) projected on silicon atoms shows lower hybridization and narrower distribution of $3s/3p$ states for Zintl Li-Si and NaSi alloys than that of pure Si, suggesting that significantly weaker Si-Si bond strength in Zintl Li-Si/NaSi alloys (Supplementary Fig. 1), making it more favorable for cleavage upon contact with solvent molecules. Furthermore, the crystal structures reveal that Zintl alkalis-Si alloys consist of

discrete small Si fragments (Fig. 1a and Supplementary Fig. 2a), in contrast to the tetrahedral network structure observed in pure Si. Specifically, $Li_{12}Si_5$ contains isolated Si atoms, $Li_{13}Si_4$ comprises a combination of $Si_2$ dumbbells and isolated Si atoms, $Li_7Si_3$ is primarily composed of $Si_2$ dumbbells, $Li_{12}Si_7$ features $Si_4$ stars-shaped structures and $Si_5$ rings, and NaSi is constructed from $Si_4$ tetrahedrons. Moreover, the electron localization functions (ELF) reveal that, in addition to the liberated free valence electrons, electrons transferred from alkali atoms to Si atoms are significantly localized at the outer shells of Si sites, promoting the formation of unpaired electrons (Fig. 1a and Supplementary Fig. 2b). The Bader charge analysis (Supplementary Table 1) shows that the mean charge state of Si atoms in $Li_{12}Si_7$ (−1.40), $Li_7Si$ (−1.88), $Li_{13}Si_4$ (−2.62) and $Li_{21}Si_5$ (−3.36), NaSi (−0.76) are generally more negative than that of pristine Si ( ~ 0). Meanwhile, the charge states of Li atoms remain relatively stable ( + 0.80 ~ +0.82) and those of Na atoms are +0.76. Furthermore, the electron states of various Si clusters show a decrease from 0 to −3.36 with increasing alkalis atom content (Supplementary Table 2). Smaller Si clusters containing more unpaired electrons offer greater potential for facilitating the rapid activation of water/methanol during the hydrogen production process. Correspondingly, within the same Zintl $Li_{13}Si_4$ alloy, the Si single atom site with a charge state of −3.16 demonstrates superior competitiveness than that of the Si dumbbells site ( − 2.07) in triggering solvent molecules dissociation.

To further verify the results obtained from the theoretical calculation, four Li-Si ($Li_{12}Si_7$, $Li_7Si_3$, $Li_{13}Si_4$ and $Li_{21}Si_5$) and one NaSi alloy were synthesized, while the nano Si sized at approximately 50 nm was used as a comparison (Supplementary Fig. 3a). The X-ray diffraction (XRD) patterns and scanning electron microscope (SEM) images revealed that the four Li-Si alloys show single-phase structures with similar microstructures, although slight oxidation was observed in the $Li_{21}Si_5$ alloy (Fig. 1b and Supplementary Figs. 3b–e). The NaSi alloy consists of NaSi phase along with minor Si phases (Supplementary Fig. 4). Moreover, electron spin resonance (ESR) spectra revealed a new signal at g = 2.003 for Zintl Li-Si and NaSi alloys, while no signal was detected for the nano Si (Fig. 1c and Supplementary Fig. 5). This signal is ascribed to the unpaired electrons localized at Si-deficit sites within the Zintl phase structure, resulting from the electron transfer from Li/Na atoms to Si atoms. Moreover, the intensity of ESR signal increases along with the number of Li electron donors, confirming the enhancement of the unpaired electron numbers. This finding was further supported by a gradual downshift in Si $2p$ X-ray photoelectron spectra (XPS) peaks from ~ 98.9 eV ($Si^0$) to ~ 97.2 eV (alkalis-Si)[34–36] (Fig. 1d and Supplementary Fig. 6a). Meantime, the binding energy of the Li-Si peaks ( ~ 53.9 eV) in the Li $1s$ spectra and the Na-Si peak ( ~ 1070.6 eV) in Na $1s$ spectra are both higher than those of elemental Li ( ~ 52.4 eV) and Na ( ~ 1068.0 eV)[37,38] (Supplementary Figs. 6b and 7), further validating the electron transfer from alkalis metal sites to silicon sites. The other peaks corresponding to $Si^{4+}$ for nano Si and $Si^{x+}$ for Zintl phase alloys, as well as $Li^+$/ $Na^+$ in the Li/Na $1s$ spectra, can be attributed to the inevitable surface oxidation[39].

### Hydrogen evolution performance of designed Zintl Si-based phase alloys
The hydrogen evolution of as-prepared Zintl Si-based alloys has been further explored under the condition of a 100:1 mass ratio of solvent to alloy. As shown in Fig. 2a, c, the alloys of $Li_{12}Si_7$, $Li_7Si_3$ and $Li_{13}Si_4$ achieve high hydrogen yields of 1.059 L g⁻¹, 1.539 L g⁻¹ and 1.595 L g⁻¹ in pure water within 10 mins at 25 °C, respectively. Notably, the $Li_{21}Si_5$ alloy produces a hydrogen yield of 1.643 L g⁻¹ within just 1 min. Additionally, the maximal hydrogen production for the $Li_{12}Si_7$ alloy achieves 1.334 L g⁻¹ after 80 min (Supplementary Fig. 8). When considering the hydrogen yield alongside hydrolysis rate, the Zintl $Li_7Si_3$ alloy emerges as the superior option under pure water conditions. Moreover, the Zintl $Li_{21}Si_5$ alloy demonstrates an unprecedentedly

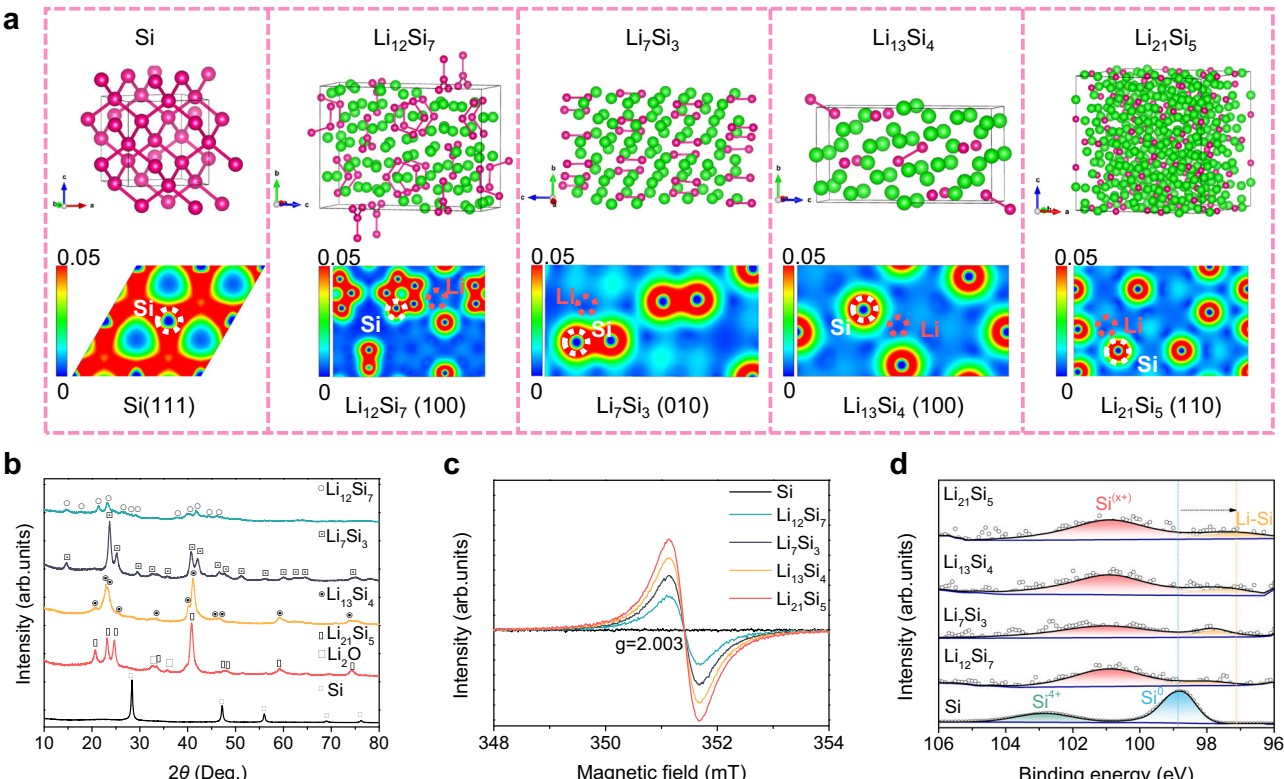

**Fig. 1 | Characterization of Zintil Li-Si alloys and nano Si. a** Crystal structure and calculated ELF results. Red to blue suggests reduced electron localization. **b** XRD patterns. **c** ESR spectra, in which the signal at g = 2.003 corresponds to the electrons trapped at a Si-deficit due to the electron transfer from Li atoms to Si atoms. **d** XPS spectra of Si 2*p* element.

high hydrogen yield of 1.739 L g$^{-1}$ in methanol within only 0.6 min. Meanwhile, the Li$_{12}$Si$_7$, Li$_7$Si$_3$ and Li$_{13}$Si$_4$ alloys respectively liberate 0.849, 0.988 and 1.616 L g$^{-1}$ H$_2$ in methanol within 10 mins. In contrast, the nano Si powder exhibited no reactivity towards either pure water and methanol. Moreover, it can be observed that the hydrogen production process of Li$_{13}$Si$_4$, Li$_7$Si$_3$ and Li$_{12}$Si$_7$ alloys in water exhibits slow hydrogen kinetics after the initial rapid hydrogen evolution. However, this phenomenon is less pronounced in methanolysis systems. The observed kinetic disparity primarily stems from the different catalytic effects of the concomitant alkali metals' products. Specifically, LiOH continues to catalyze the reaction between residual silicon and water, sustaining hydrogen generation until complete passivation of the silicon species occurs. In contrast, CH$_3$OLi shows no catalytic activity toward the silicon-methanol reaction, which has been further demonstrated in the Supplementary Figs. 19–22 (which will be discussed later).

The maximal Si utilization rate of the alloys in the two systems is calculated based on the "hydrogen yield after deducting the hydrogen production of Li / theoretical hydrogen yield of Si species" (Supplementary Table 3). As presented in Fig. 2b, d, the Zintl Li$_{21}$Si$_5$ alloy delivers ultra-high maximal Si utilization rates of 86.9% in pure water and 98.1% in methanol. We conduct a thorough comparison of the developed hydrogen production systems with previously reported Si-based hydrogen production systems. Notably, the Zintl Li$_{21}$Si$_5$ alloy-H$_2$O/CH$_3$OH demonstrates remarkable advancements in terms of materials synthesis efficiency, reaction solution safety, total hydrogen yield, average hydrogen evolution rate and Si utilization rate (Supplementary Fig. 9 and Table 4). It is noteworthy that no studies have demonstrated the feasibility of effectively conducting the Si-methanol reactions at a temperature below 25 °C[40]. Moreover, the as-prepared NaSi powder also achieves hydrogen yields of 0.911 L g$^{-1}$ in pure water and 0.866 L g$^{-1}$ in methanol within 3 min, corresponding to the Si

utilization rates of 68.0% and 63.5% (Supplementary Fig. 10), respectively. Additionally, the maximal Si utilization rates either in water or methanol increase with the transition of the Zintl phase structure from Li$_2$Si$_7$ to Li$_{21}$Si$_5$. The trend is consistent with the changes in the bonding structure and electron state of the Si atom across various Zintl phase alloys, where the Zintl Li$_{21}$Si$_5$ alloy containing the most discrete Si atoms and enriched unpaired electrons exhibits superior hydrogen generation performance compared to others. Notably, the yielded H$_2$ from both two systems is highly pure without byproducts such as SiH$_4$ and Si$_2$H$_6$, except for trace amounts of H$_2$O vapor and methanol vapor, as confirmed by the mass spectrometry (MS) analysis (Supplementary Fig. 11). Furthermore, the exothermicity investigations of Zintl Li-Si alloys' hydrogen production systems show their vigorously exothermic properties during the operation process (Supplementary Figs. 12 and 13), while the hydrogen production behaviors of Li$_{21}$Si$_5$ alloy conducted within various amount of pure water highlights the crucial role of the amount of solvent in controlling both the temperature of reaction system and the hydrogen evolution kinetics (Supplementary Fig. 14).

Motivated by the exceptional hydrogen generation properties of the Zintl Li$_7$Si$_3$ alloy-water system and Zintl Li$_{21}$Si$_5$ alloy-methanol system at room temperature, we further investigated the hydrogen evolution kinetics and corresponding Si utilization rates of both systems at various temperatures (10, 20, 30 and 40 °C) to assess their suitability for regular outdoor applications. As illustrated in Fig. 2e–g, the Zintl Li$_7$Si$_3$ alloy-water system exhibits a splendid hydrogen production/Si utilization rate of 1.396 L g$^{-1}$/67.8% within 10 min at 10 °C, which gradually increases to 1.566 L g$^{-1}$/83.2% at 40 °C. Whereas, the Zintl Li$_{21}$Si$_5$-methanol system shows a slight impact on the hydrogen production performance when the temperature varies, with the hydrogen yield and utilization rate in 1 min increasing from 1.656 L g$^{-1}$/88.4% (10 °C) to 1.754 L g$^{-1}$/99.9% (40 °C). These results suggest that

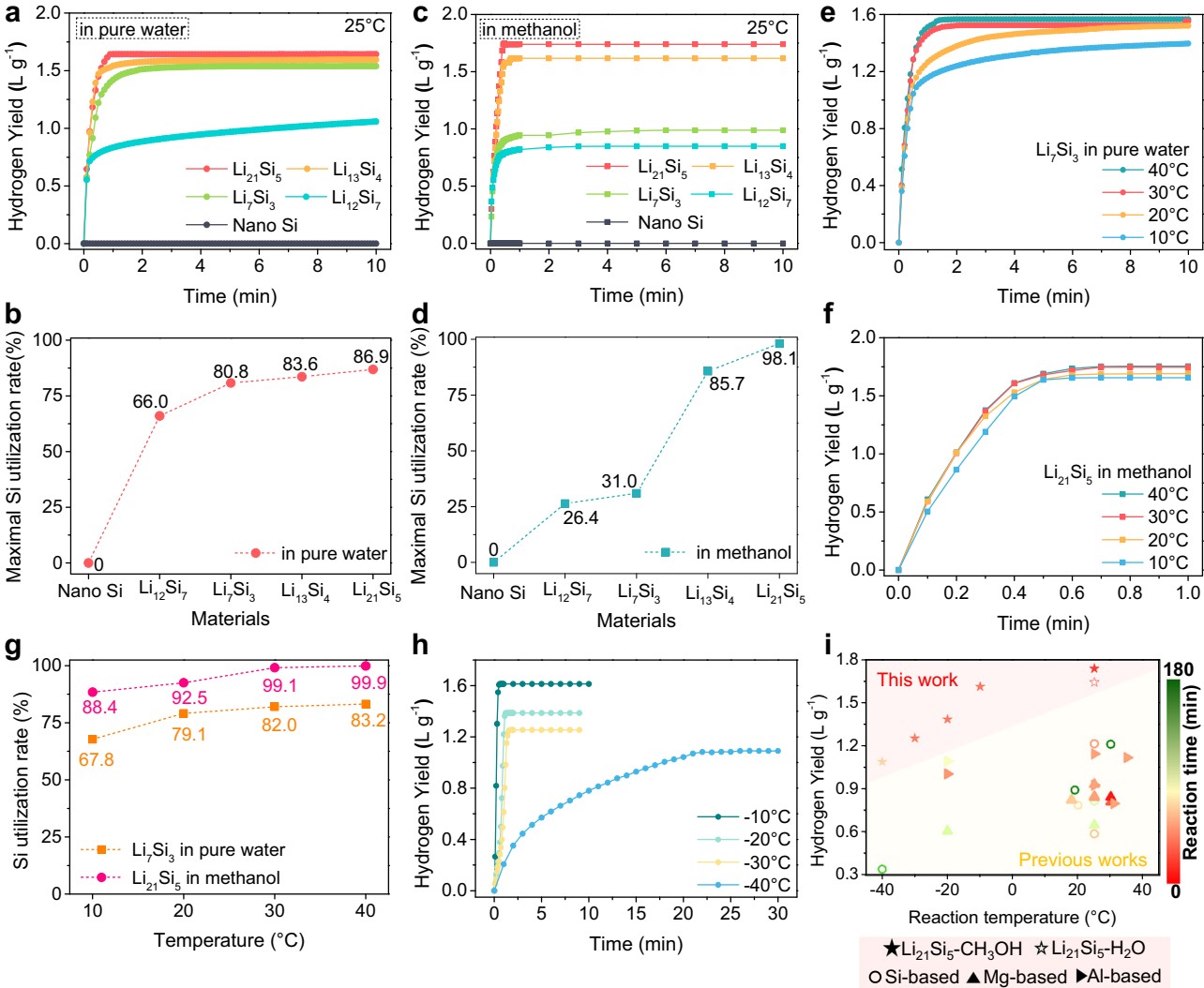

**Fig. 2 | Hydrogen evolution properties of Zintil Li-Si alloys and nano Si.** Hydrogen evolution curves and related maximal Si utilization rates of Zintl Li-Si alloys and nano Si in (**a**, **b**) pure water/ (**c**, **d**) methanol at 25 °C, respectively. Hydrogen evolution curves of (**e**) Zintl $Li_7Si_3$ alloy in pure water and (**f**) Zintl $Li_{21}Si_5$ alloy in methanol at different temperatures, and (**g**) the corresponding Si utilization rates. **h** Hydrogen evolution curves of Zintl $Li_{21}Si_5$-methanol system at different subzero temperatures. **i** Comparison of hydrogen evolution performance of Zintl $Li_{21}Si_5$ alloy with the state-of-the-art Si/Mg/Al-based materials proceeded in the neutral or nearly neutral solutions.

both systems are well-suited for practical outdoor applications. Furthermore, the hydrogen evolution ability of Zintl $Li_{21}Si_5$ in methanol was further evaluated under even lower temperatures, and the results are illustrated in Fig. 2h. It shows that the Zintl $Li_{21}Si_5$-methanol system could release 1.613, 1.386 and 1.254 L g$^{-1}$ $H_2$ within less 2 min at −10 °C, −20 °C, −30 °C, respectively. Even at −40 °C, 1.091 L g$^{-1}$ of hydrogen gas is generated after 26 min, far surpassing the previous studies (Fig. 2i and Supplementary Table 5). Therefore, these considerable hydrogen production results emphasize that as-prepared Zintl phase alloys can enable excellent hydrogen supply in a fairly wide temperature range, especially for the ultra-low temperature environment below 0 °C, with significant implications for distributed hydrogen utilization. Moreover, we further developed a paraffin (PA) coating strategy for improving the air stability of $Li_{21}Si_5$ alloy (Supplementary Fig. 15). The super-hydrophobic paraffin coating can effectively mitigate moisture-induced degradation of the internal alloy by acting as a barrier against humid air. As a result, the as-prepared $Li_{21}Si_5$@PA achieves an excellent hydrogen yield retention of 96.6% after 1 h humid air exposure (64–70% humidity, -28 °C), much superior to the $Li_{21}Si_5$ alloy of 58.4% (Supplementary Fig. 16).

## Hydrogen release mechanism

To elucidate the underlying mechanism governing the enhanced Si conversion efficiency and kinetics observed in Zintl phase alloys, in situ FTIR spectroscopy measurements were conducted to capture the transient reaction intermediates during two hydrogen evolution systems. The Zintl $Li_{21}Si_5$ alloy was employed as the substrate due to its abundance of active Si sites among these substrates. As illustrated in Fig. 3a, a prominent peak is observed at approximately 650 cm$^{-1}$, which is related to the Si-H group[29], with its intensity gradually diminishing over the course of the reaction. Concurrently, an increase in the intensity of the Si-O-Si peak at 694 cm$^{-1}$ in $SiO_2$ sol is observed, indicating the formation of silicon-oxygen bonds. The emergence of the Si-H bond serves as compelling evidence for the electron transfer from the negatively charged Si species of $Li_{21}Si_5$ to $H_2O$ molecules. Subsequent nucleophilic attacks by other $H_2O$ molecules on the Si-H bonds lead to the formation of Si-OH and the release of $H_2$ gas. The Si-OH intermediates then rapidly undergo self-polymerization or react with LiOH, generating the Si-O-Si structure[32]. Analogous behavior is observed during the methanolysis process, with the detection of Si-H structures via IR peaks at around 645 cm$^{-1}$ and 743 cm$^{-1}$, as shown in

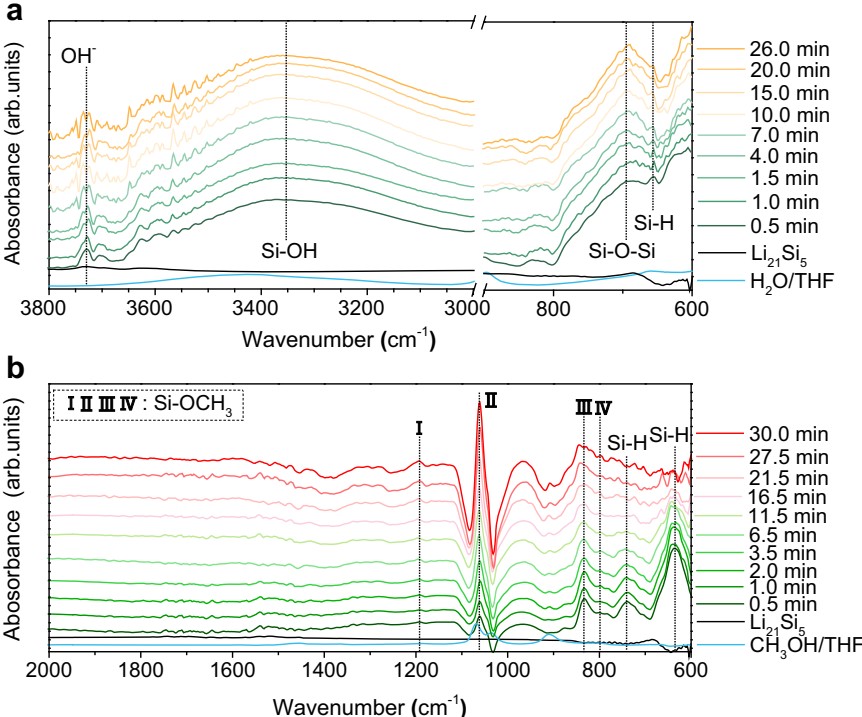

**Fig. 3 | In situ FTIR spectroscopy characterization.** In situ FTIR analysis of Zintl Li$_{21}$Si$_5$ alloy (**a**) hydrolysis and (**b**) methanolysis process.

Fig. 3b. The yielded Si-H groups are further nucleophilic attacked by CH$_3$OH molecules, resulting in the gradual disappearance of Si-H bonds and the enhancement of Si-OCH$_3$ radical[41]. It indicates that Si species in Zintl phase harboring significant unpaired electrons are capable of engaging in remarkable electron interactions with H$_2$O and methanol molecules, which is further affirmed by the characterization of liquid reaction products (Supplementary Figs. 17, 18). Moreover, the control experiments of nano Si in the alkaline media also highlight the importance of bonding configuration and electronic state of Si on hydrogen production performance (Supplementary Fig. 19). This is further confirmed by the hydrogen production tests of the designed H-Li$_{21}$Si$_5$ composite in Supplementary Figs 20–22. As shown in Supplementary Fig. 20, the H-Li$_{21}$Si$_5$ composite contains fresh elemental Si dispersed within the LiH matrix through the hydriding of Zintl Li$_{21}$Si$_5$ alloy. It can exclude the passivation effect of native surface oxide on the hydrogen evolution of Si species, while the LiH can also produce LiOH and CH$_3$OLi solution during the hydrolysis/methanolysis. As a result, the H-Li$_{21}$Si$_5$ shows Si utilization rates of 39.3% and 0% in water and methanol, respectively, markedly lower than those of Li$_{21}$Si$_5$ alloy (Supplementary Fig. 21). Especially, in the case of H-Li$_{21}$Si$_5$ methanolysis, the IR absorption peaks attributed to the Si-O-C structure of tetramethoxysilane (TMOS) are absent for its liquid product, corroborating the absence of a Si-CH$_3$OH reaction (Supplementary Fig. 22).

DFT calculations were performed to further elucidate the disparities in the configurations of Zintl phase alloys to pure Si when interacting with H$_2$O/CH$_3$OH solvents during the activation step. As exhibited in Fig. 4a–d and Supplementary Figs. 23, 24, the pure Si exhibits a feeble attraction to the H atom of OH* in H$_2$O/CH$_3$OH, whereas the electron-rich Si atoms in Zintl phase alloys display heightened hybridization with the H atom, suggesting their efficacy in activating solvent molecules to yield active intermediates enriched with Si-H structures. Moreover, pure Si exhibits high energy barriers of 1.47 eV and 1.24 eV for H$_2$O/CH$_3$OH dissociation (Supplementary Fig. 25, Fig. 4e, f), respectively, consistent with their non-reactivity reflected in the experiments. Whereas the energy barriers for the reactions of Zintl Li-Si alloys with water and methanol decrease

dramatically, indicating that unpaired electronic structure between Si and Li effectively promotes electron interaction with H$_2$O and CH$_3$OH molecules (Supplementary Figs. 26–30, Fig. 4e, f). These findings are further verified by energy barriers of 0.44 eV and 0.64 eV for hydrolysis and methanolysis of Zintl NaSi alloy, respectively (Supplementary Figs. 31, 32). Notably, the Zintl Li$_{21}$Si$_5$ alloy, featured by a single Si atom structure, exhibits ultra-low activation barriers of −1.13 eV and −1.47 eV for H$_2$O and CH$_3$OH, respectively, indicative of exceptionally intense electron interactions with solvent molecules to spontaneously break the H-O bond of H$_2$O or CH$_3$OH molecules during the hydrogen evolution process. Intriguingly, even within the same Zintl Li$_{13}$Si$_4$ alloy, the activation barrier energy for H$_2$O and CH$_3$OH at the Li$_{13}$Si$_4$-2Si site (0.06 eV and 0.03 eV) is higher than the Li$_{13}$Si$_4$-1Si site (−0.69 eV and −1.13 eV), reinforcing the significance of the degree of Si structure dissociation. It is notable that the reductions in activation barrier energy ($\triangle G_{MA}$) show strong linear correlations with the number of electrons transferred between Li and Si, irrespective of whether the reaction occurs in H$_2$O or CH$_3$OH (Fig. 4g). This trend further underscores that the intrinsic electronic transfer effect between Li and Si dominates the activation kinetics of solvent decomposition. Moreover, the robust linear $\Delta G_{MA}$–electron transfer correlations provide a quantifiable descriptor for rational alloy design and reaction kinetics adjustment based on various practical hydrogen generation demands.

According to the results obtained from the aforementioned experimental studies and theoretical simulations, a proposed reaction mechanism is summarized in Fig. 4h. The Zintl Si-based alloys feature well-defined unpaired electrons situated between the alkali metal atoms and Si atoms. The solvent molecules adsorb on the surface of the alloy and conduct fast structural rearrangements. Then, the high-activity unpaired electrons of alloys enable effective electronic interaction with absorbed solvent molecules, facilitating the cleavage of H-O bond in hydroxide radicals. This interaction subsequently leads to the formation of a (RO-alkalis)−SiH intermediate. Finally, the spontaneous reaction between the (RO-alkalis)−SiH intermediate and subsequent solvent molecules culminates in the efficient release of hydrogen gas.

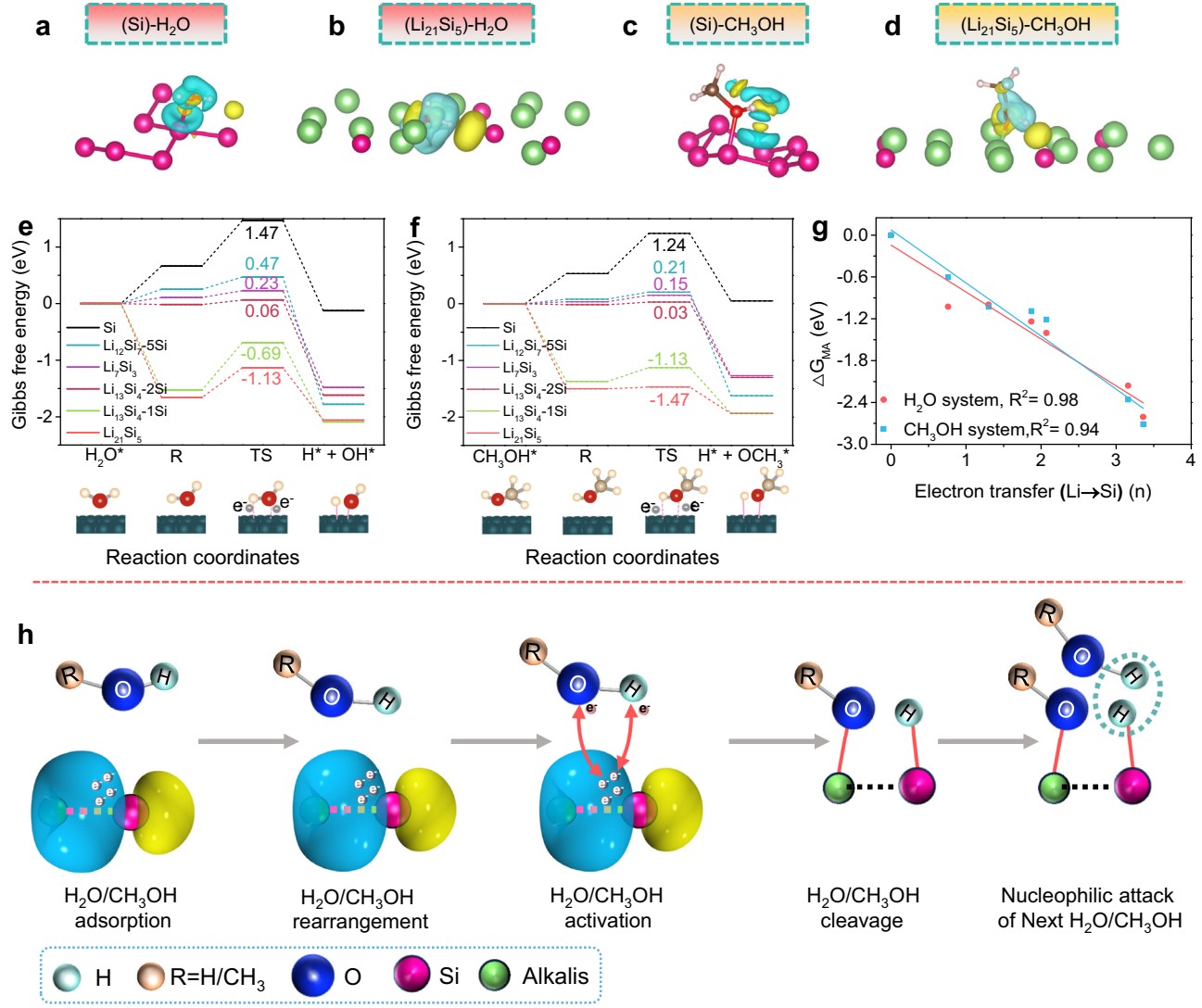

**Fig. 4 | DFT calculations. a–d** Difference density of electron distribution of configurations for the $H_2O$/$CH_3OH$ activation step, while the yellow and blue isosurfaces are attributed to the enhancement of electrons and the depletion zone, respectively. Energy profiles of (**e**) $H_2O$ and (**f**) $CH_3OH$ dissociation process on Si and various Zintl Li-Si alloy surfaces. **g** Linear relationships between electron transfer number in Li-Si alloys and $\triangle G_{MA}$. **h** Schematic illustration of $H_2O$/$CH_3OH$ dissociation on the Zintl Si-based phase alloys.

## Green recycling avenue for the lithium and anode of degraded LIBs

Zintl Li-Si phases also are the charged products of Si-based anode for the LIBs. Therefore, the advantages of effective Zintl Li-Si phase/water reaction and soluble products exhibit promise in separating the Si and graphite from the degraded graphite-silicon anodes. Here, we adopted an eco-friendly "charge-hydrolysis-separation" approach to selectively recover lithium, graphite, Si sources and Cu foil from the spent batteries under a mild condition, without pollution and toxic gas emissions (as shown in the Method section). It can help increase the earnings of battery recycling process, particularly as the graphite has been considered as a "future critical material" alongside Li and Co[42]. As shown in Fig. 5a, the residual active lithium of the LIBs was concentrated in the anode through a CC-0.05 C/CV-0.005 C charging process. Subsequently, the disassembled anode was processed with distilled water (Supplementary Movie 1). The dissolution of Li-Si phase and the lithium from the $Li_xC$ to water proceeded spontaneously by the corresponding hydrogen evolution reactions at room temperature within less than 1 min. Concurrently, the intact Cu foil separates from anode materials for recovery (Supplementary Movie 1 and Fig. 5b). High purity graphite and concomitant graphene are then obtained

through a simple filtration (Fig. 5b, c and Supplementary Figs. 33, 34), which could be further used for energy storage or other functional applications[42]. The Li and Si in the residual solution can be completely recovered in the forms of LiCl and amorphous $SiO_2$ (Fig. 5b and c) via a simple neutralized-redissolution process. Importantly, the alkaline nature of the solution neutralizes HF and $H_3PO_4$ species yielded from the hydrolysis of electrolyte and the solid electrolyte interphase (SEI) layer of anode[43], thereby preventing the production of toxic waste.

We further compared the developed recycling process with the presented mainstream LIBs recovery technologies of pyrometallurgy (pyro-) or hydrometallurgy (hydro-) methods. As shown in Fig. 5d, either pyrometallurgy or hydrometallurgy methods to recover valuable materials are conducted on a fully discharged state of the end-of-life LIBs[44]. However, on the one hand, the graphite becomes $CO_2$ footprint and the lithium and Si would be discarded in the effluent slag after the pyrometallurgical processes[45,46]. On the other hand, the high leaching temperature (>70 °C) for hydrometallurgy leads to the emission of toxic gases like $Cl_2$ and $NO_x$, and the extraction of high-purity Li salt is complex. Moreover, while the established anode recycling methods have developed the recovery of active material and Li resources from the pure Sn anode[47] or commercial graphite anode[48],

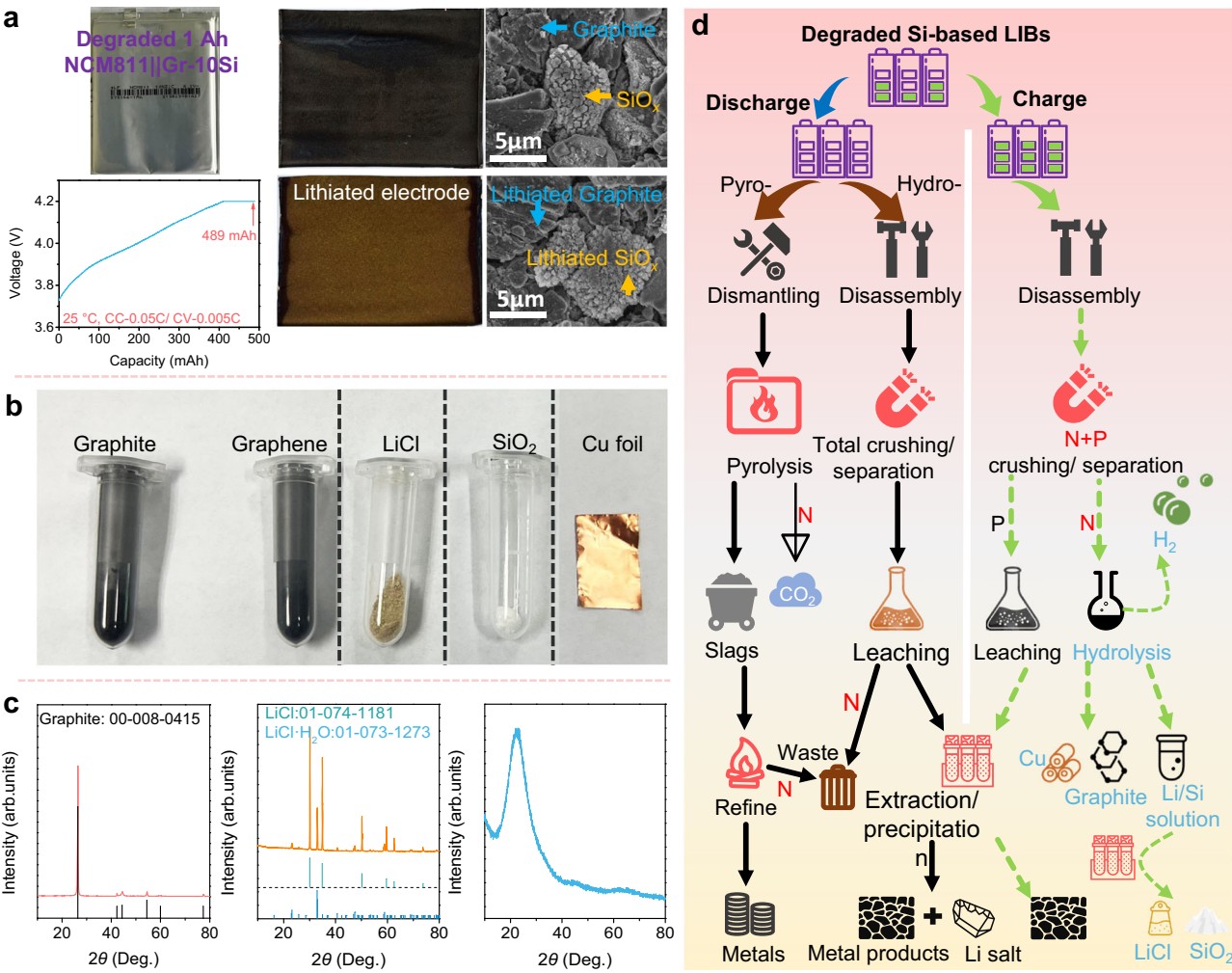

**Fig. 5 | Recycling of lithium and anode materials of degraded LIBs. a** The characterizations of pristine and charged degraded 1 Ah NCM 811 || Gr-10Si pouch cell. **b** The photos of recovered graphite, graphene, LiCl, SiO₂ and Cu foil. **c** XRD patterns of recovered graphite, LiCl and SiO₂. **d** Comparison of the pyro-, hydro- and as-developed recycling technologies.

these methods face challenges in separating Si and graphite components from spent next-generation Si-based LIBs. Moreover, the conventional recovery strategies of graphite anode often necessitate the time-consuming leaching or pyrolysis to remove the binders between active materials and current collectors. However, leaching requires additional chemical regents, while pyrolysis increases energy consumption and produces a lot of byproducts such as pyrolysis oil and hazardous gases (e.g., HF, dioxins and fluorinated organic compounds), further escalating the cost of anode recovery and posing possible risks of environmental pollution. In contrast, the proposed green recovery method not only can work out at room temperature without the emission of carbon footprint and poisonous gas, but also is beneficial for feasibly realizing the recovery of multifarious anodic materials, especially for high-purity Li salt, graphite and Si resource.

## Discussion

We have demonstrated that the internal electron transfer within the Zintl Si-based phase promotes the valence electron release of Si from its covalent bonds, leading to the formation of free unpaired electrons with electrons donated from other sources. This results in effective electron interactions towards water/methanol with significantly reduced barriers, facilitating the generation of Si-H rich intermediates that promote hydrogen evolution. Consequently, the designed Zintl alkalis-Si alloys exhibit desirable hydrogen production yields and Si

utilization rates at room temperature and ultra-low temperature. In particular, Zintl $Li_{21}Si_5$ alloy releases 1.643 and 1.739 L g$^{-1}$ H₂ in water and methanol at room temperature, respectively, corresponding to the Si utilization rate of 86.9% and 98.1%, far exceeding that of previous reports. Moreover, the as-prepared alloy unprecedentedly achieves fast hydrogen conversion in pure methanol at −40 °C and produces 1.091 L g$^{-1}$ H₂ within 26 min. This work provides an attractive approach to expanding the applications of Si-based hydrogen production, particularly for portable devices or challenging outdoor environments such as polar areas and plateau regions. Furthermore, the fundamental understanding of Zintl Li-Si phase-water reaction promotes the development of a green recovery route for the lithium and anode materials of degraded LIBs.

## Methods

### Materials and chemicals

Li metal (99%) was purchased from China Energy Lithium Co., Ltd. Na metal (99%), LiOH (98%), tetrahydrofuran (99%) and CH₃OH (99%) were purchased from Shanghai Aladdin Biochemical Technology Co., Ltd. NaOH (99%) and n-heptane (99%) were purchased from Shanghai Macklin Biochemical Co., Ltd. The nano Si powder (99%) was purchased from Xuzhou Jiechuang New Material Co., Ltd. The degraded NCM811|| Graphite (Gr)-10Si commercial pouch cell was purchased from Lifang new energy Technology Co., LTD. CH₃OLi solution and

CH₃ONa solution were homemade through the reactions between methanol and Li/Na metals.

## Preparation of materials

**Li-Si alloys preparation.** A mixture of Li sheets and Si powder with Li/Si ratio of 21:5 was heated at 250°C in an argon-filled glovebox, inducing the liquefaction of lithium and subsequent infiltration into Si powder. Then, the mixture was subjected to a continuous stirring process until achieving well distribution of Li metal across silicon powder surface, yielding the black products. Afterward, the black products were mechanically alloyed to obtain $Li_{21}Si_5$ alloy using a vibration-type miller (QM-3C, Nanjing, China) with 1200 rpm for 3 h. In the milling process, the powder was sealed into the bearing-steel vessel with bearing-steel balls in a mass ratio of 50:1 and 3 mL n-heptane as the dispersant. In addition, the residual n-heptane was effectively removed by heating the milled products in the glove box at 80 °C for 30 min. Compared to $Li_{21}Si_5$, the synthesis of $Li_{12}Si_7$, $Li_7Si_3$ and $Li_{13}Si_4$ only adjusted the atomic ratio of Li and Si.

**NaSi alloy preparation.** The mixture of Na sheet and Si powder with Na/Si ratio of 1:1 was pre-wrapped in a Ti foil, and heated at 400 °C for 24 h by the vacuum-sealed tube sintering.

**Li₂₁Si₅@PA preparation.** 0.005 g paraffin (PA) was dissolved in 1 mL n-heptane and the mixture was then applied dropwise to cover the entire surface of a 0.1 g $Li_{21}Si_5$ tablet. The PA can spontaneously precipitate upon the rapid evaporation of n-heptane solvent, thus forming a uniform PA coating film on the surface of alloy tablet within 10 s.

## Characterization

XRD pattern was recorded by the X-ray diffractometer (PANalytical EMPYREAN) with CuKα radiation (45KV, 40 mA). ESR spectra were tested by Bruker A300 electron spin resonance spectrometer. SEM and EDS images were conducted on the scanning electron microscope (Zeiss Supra 40/VP). A thermal camera (RSE60 Fluke) was used to record the temperature change in the hydrogen production process. ATR-FTIR (Nicolet IS50) and GC-MS (QP2010ultra) were employed to analyze the chemical structure of reaction products. The gas generated from the hydrogen evolution process was detected using a Hiden-Qic 20 mass spectrometer.

As for in situ FTIR tests, because the reactions of Zintl $Li_{21}Si_5$ alloy in pure water and methanol are too fierce to detect intermediates, the H₂O/THF (V/V = 1:6) and methanol/THF (V/V = 1:6) mixtures were used as reaction reagents, where the THF just acts as a diluent to reduce the reaction rate.

As for the characterization of the liquid products of the hydrolysis and methanolysis systems, the hydrolysis liquid products were dried at 80 °C and analyzed using XRD, while the methanolysis liquid products were detected directly using FTIR and GC-MS.

## Hydrogen evolution measurements

Before the test of hydrogen production, 0.1 g Li-Si alloy was pressed into a tablet under the pressure of 3 MPa within 15 s, to avoid the combustion of highly dispersed alloy powder in water and methanol. The nano Si powder and the as-prepared NaSi powder were used directly for the hydrogen production reaction. Then, the hydrogen production performance of alloys was carried out with the following steps. 10 ml water or methanol was injected into a Pyrex flask reactor preloaded with reaction material. The produced hydrogen was exhausted through the first Monteggia washing bottle to absorb the heat of gas and condense the water/methanol vapor, and then passed through the second Monteggia washing bottle to expel water. The expelled water was collected by a breaker put on an electronic scale connected to a computer, which was used to compute the

volume of hydrogen gas by recording the weight of the extracted water over time.

## Recycling process of degraded LIBs

The degraded NCM811||Graphite (Gr)-10Si commercial pouch cell was firstly charged to 4.2 V at a constant current (CC) of 0.05 C and followed by a constant voltage (CV) till 0.005 C. Then, the pouch cell was dismantled, and the anode/cathode plates were separated from each side of the separator. Next, the anodic electrode was loaded in a reaction tube to contact with the excess water for 1 min, following the related reactions:

$$Li_xSi_{(s)} + (x+1)H_2O_{(l)} \rightarrow (x-2)LiOH_{(aq)} + Li_2SiO_{3(aq)} + (x+4)/2H_{2(g)} \tag{1}$$

$$Li_xC_{(s)} + xH_2O_{(l)} \rightarrow xLiOH_{(aq)} + C_{(s)} + x/2\,H_{2(g)} \tag{2}$$

The Cu foil was automatically detached during the hydrogen evolution process and insoluble graphite/graphene solids could be collected through filtration. The graphene can be separated from the mixture using centrifugation.

On the other hand, the low-concentration HCl solution was introduced into the remaining solution part, therefore resulting in the production of LiCl (Equation 3) and $H_2SiO_3$ (Equation 4).

$$Li_2SiO_{3(aq)} + 2HCl_{(aq)} \rightarrow 2LiCl_{(aq)} + H_2SiO_{3(aq)} \tag{3}$$

$$LiOH_{(aq)} + HCl_{(aq)} \rightarrow LiCl_{(aq)} + H_2O_{(l)} \tag{4}$$

The $H_2SiO_3$ can be converted to $SiO_2$ solid via drying, which could be further separated with LiCl through a redissolution process.

## DFT calculation

The DFT calculations were executed employing the Vienna Ab-initio Simulation Package[49] (VASP). The van der Waals (vdW) correction was considered by applying the DFT-D3 approach[50,51]. Projector-augmented plane wave (PAW) pseudopotentials were employed, along with the generalized gradient approximation using the Perdew-Burke-Ernzerhof functional (GGA-PBE), to effectively describe the exchange-correlation function[52,53]. The Brillouin zone was sampled with Gamma (Γ)- centered Monkhorst-Pack mesh for geometry relaxation, with a K-mesh of 0.03 Å⁻¹. A cutoff energy of 520 eV was employed throughout all calculations, with structural relaxation performed via the conjugate gradient method until atomic forces converged below 0.03 eV/Å. The climbing image nudged elastic band (CI-NEB) method was used to calculate the reaction barriers[54].

## Data availability

All relevant data generated in this study are provided in the Supplementary Information/Source Data file. Source data are provided with this paper.

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

## Acknowledgements
L.Z.O.Y. acknowledges the financial support from the National Natural Science Foundation of China Projects (52271213). L.J. acknowledges the support from the National Natural Science Foundation of China Projects (52371229).

## Author contributions
M.L.L., Q.Y.J., H.L., and L.Z.O.Y. conceived and designed the project. M.L.L. conducted most of experiments with the help of J.W.L., K.C., and H.Z. Q.Y.J. performed the DFT calculation. M.L.L. prepared the manuscript with advice from L.J., H.L., L.Z.O.Y., and M.Z. L.J., H.L., L.Z.O.Y., and M.Z. provided supervision and resources for this work. All the authors discussed the data and revised the paper.

## Competing interests
The authors declare no competing interests.
