## [Transparent Peer Review file · Nature Communications]

Activating Silicon for High Hydrogen Conversion and Sustainable Anode Recovery

Corresponding Author: Professor Lin Jiang

Version 0:

Reviewer comments:

Reviewer #1

(Remarks to the Author)

This paper describes the use of the reaction of Li-Si Zintl phase materials with water and methanol to generate hydrogen. The paper is interesting but would require significant modification to be appropriate for a Nature family journal. From a cost perspective, hydrogen generation from a solid plus water (whether Si, Al, B, or other) is only ever going to be practical for very small applications where energy density is paramount. The cheapest silicon (metallurgical grade) is about \$2/kg, and it takes a minimum of 7 kg Si to make 1 kg of hydrogen. Lithium is much more expensive. Thus, the minimum possible cost of hydrogen from this process will be well above \$14/kg. This price is comparable to what I pay to have compressed gas cylinders of nitrogen delivered to my lab. Because of the limited applicability of this mode of hydrogen generation, I think the most interesting part of the article is the idea of using the reaction of lithiated silicon with water in recycling of silicon-containing anodes from lithium-ion batteries. Thus, the authors may want to shift the focus of the manuscript to emphasize this more. However, some of the same authors reported a similar process for reacting lithiated tin anodes with water to generate hydrogen (<https://doi.org/10.1016/j.jallcom.2023.169548>) which I didn't see cited in this paper. Thus, if the authors do decide to focus more on the battery recycling application, they should be sure to establish the novelty of this approach relative to all other prior reports.

Figure 1(b), as only a qualitative illustration, doesn't add much to the article. It might be more appropriate as part of a TOC figure. The same is true of Figure 3(i).

A very important control experiment, which was not reported, would be to test hydrogen generation by Si nanoparticles with addition of lithium hydroxide or sodium hydroxide equivalent to the lithium or sodium content in the Zintl phase. This would rule out the possibility that lithium hydroxide is simply catalyzing reaction of Si.

What are the exothermicities of the various reactions and how much heating is observed? The authors should at least mention this, as managing the exothermicity would be critical at larger scale.

The authors use absorbance at 645-650 wavenumbers as evidence of Si-H bonds, but we would usually identify Si-H based on stretching modes near 2200 wavenumbers. Were those peaks evident? Or is that spectral range inaccessible for some reason?

The authors should more clearly present the fraction of hydrogen generation attributable to Li and to Si in the main manuscript. I was somewhat confused by Figure S15(c). The authors should clarify what they mean by "total" in that figure or label the two parts of each bar as Li and Si. For practical purposes, comparisons on a mass basis are more relevant than on a molar basis (and a mass basis is used elsewhere).

The authors should further explain the negative activation barriers that they report – how are these "barriers" if they are negative. Is the structure a local maximum on the reaction path? If so, there should be a local minimum between the reactions and this nominal transition state structure. Adding the local minimum to the energy diagrams (Fig. 5e-f) would be helpful.

I noticed that the results of refs. 18 and 19 of the main manuscript were not included in the comparison table in the SI (table S4). The authors should ensure they have included comparisons to all prior reports if they want to claim to have

outperformed all prior methods. Ref. 18 focuses mainly on the speed of hydrogen generation but also claims to produce more than 2.0 moles of dihydrogen per mole of silicon, attributing the excess to the hydrogenated surface of nanoparticles. This would imply 100% silicon utilization. Similarly, ref. 18 focuses on a potential biomedical application, but seems to claim 100% Si utilization, with > 2 moles dihydrogen per mol Si, again attributable to hydrogenated surfaces.

The methods section is inadequate. For example, "mixture of Li sheet and Si powder with Li/Si ratio of 21:5 was heated and continuously stirred to become well-distributed." Heated to what temperature? Stirred how? In what form were the FTIR measurements made? The authors should provide a level of detail sufficient for someone to reproduce their work.

The authors should clearly indicate in the results section the quantity of excess water or methanol being used (e.g., a 100:1 mass ratio of water to solids). Does the process require this much excess water? Did they do any experiments at lesser excess water?

The authors should address the air stability (particularly with significant humidity) and hazards of the lithium-silicon materials. They may want to consult an SDS for a commercial lithium-silicon powder used in batteries ([https://www.eaglepicher.com/sites/default/files/EHS-SDS-1005 Lithium Silicon powder SDS rev Orig 5-10-18.pdf](https://www.eaglepicher.com/sites/default/files/EHS-SDS-1005%20Lithium%20Silicon%20powder%20SDS%20rev%20Orig%205-10-18.pdf)).

Reviewer #2

(Remarks to the Author)

This work reports the Zintl phase Si-based alloys featuring discrete Si clusters with substantial unpaired electrons, demonstrating superior hydrogen production yields and enhanced Si utilization efficiency in water/methanol systems across a broad operational temperature range. The prominent unpaired electron configuration in these alloys achieves exceptional interaction with solvent molecules, thereby remarkably reducing the activation barriers of H₂O/CH₃OH dissociation as evidenced by DFT calculation and in situ FTIR tests, which primarily accounts for the hydrogen production performance and Si utilization rate surpassing those reported in previous studies. This methodology carries significant implications for designing high-performance Si-based hydrogen production systems, particularly given its straightforward material preparation and environmentally benign reaction conditions. Furthermore, it provides novel insights into the sustainable recovery of valuable resources from spent LIBs. This work is well-presented and thoroughly analyzed, which is a very interesting and high-quality study from which the portable hydrogen supply technology will benefit. I recommend its publication after addressing the following minor revisions:

1. In Figure 3a and c, the hydrogen production process of Li₁₃Si₄, Li₇Si₃ and Li₁₂Si₇ alloys in water exhibit slow hydrogen kinetics after the initial rapid hydrogen evolution, however, this phenomenon is less pronounced in methanolysis systems. Could the authors elaborate on the potential origins of this solvent-dependent kinetic behavior?
2. In Fig.5g, the Δ GMA shows a linear correlation with the number of the charge transferred in the Li-Si alloys. The authors should offer additional physical insights to give the readers a better understanding.
3. While Li-Si and NaSi alloys were employed as models to investigate the role of unpaired electron structures in Si-H₂ conversion, could this strategy be rationally extended to other Si-based alloy systems?
4. Abbreviations such as "TMOS" (line 259, page 11) must be explicitly defined upon their first occurrence. Please ensure consistent adherence to this convention throughout the manuscript.
5. The detailed information regarding the chemicals and materials used should be provided.
6. In the Method section, the authors are encouraged to provide additional details regarding the sample preparation, such as the specific method for removing residual n-heptane following the ball milling process.

Version 1:

Reviewer comments:

Reviewer #1

(Remarks to the Author)

The authors have, for the most part, addressed my concerns with the original version of the paper. Their explanations in the response to the reviews are quite helpful, but in most cases the changes in the manuscript itself were minor. For example, the abstract was unchanged and still leads off with a statement about "cost-effective" hydrogen generation. I still believe that hydrogen generation will only be cost-effective in a few niche applications. The value proposition, if there is one, will be in the battery recycling application, where the value will come from recovery of the Si, Li, and carbon-based components and from reduced waste generation.

That being said, with the open peer review model of Nature Communications, I think the paper is acceptable for publication. A sufficiently-interested reader could dig into the peer review file for the detailed discussion provided in the response to reviewers.

Reviewer #2

(Remarks to the Author)

All of the issues have been well addressed in the revised manuscript, and I think the manuscript can be accepted for publication now.

Response to the reviewers' comments

Dear Reviewers,

We highly appreciate your constructive suggestions and comments, which are very helpful for improving the quality of our manuscript. We have carefully revised our manuscript according to your suggestions and comments. The point-to-point replies are as follows.

Reviewer #1 (Remarks to the Author):

This paper describes the use of the reaction of Li-Si Zintl phase materials with water and methanol to generate hydrogen. The paper is interesting but would require significant modification to be appropriate for a Nature family journal. From a cost perspective, hydrogen generation from a solid plus water (whether Si, Al, B, or other) is only ever going to be practical for very small applications where energy density is paramount. The cheapest silicon (metallurgical grade) is about \$2/kg, and it takes a minimum of 7 kg Si to make 1 kg of hydrogen. Lithium is much more expensive. Thus, the minimum possible cost of hydrogen from this process will be well above \$14/kg. This price is comparable to what I pay to have compressed gas cylinders of nitrogen delivered to my lab. Because of the limited applicability of this mode of hydrogen generation, I think the most interesting part of the article is the idea of using the reaction of lithiated silicon with water in recycling of silicon-containing anodes from lithium-ion batteries. Thus, the authors may want to shift the focus of the manuscript to emphasize this more. However, some of the same authors reported a similar process for reacting lithiated tin anodes with water to generate hydrogen (<https://doi.org/10.1016/j.jallcom.2023.169548>) which I didn't see cited in this paper. Thus, if the authors do decide to focus more on the battery recycling application, they should be sure to establish the novelty of this approach relative to all other prior reports.

Response: We appreciate the reviewer for the approval of our work and insightful suggestions to help us improve the quality of our manuscript.

We recognize that the cost factor is critical for the popularization of Si-based hydrogen production systems and have rephrased the related advantages of Si in the introduction of the revised manuscript to lower the tone. Meanwhile, we would like to highlight the high-performance of hydrogen production even at an extremely low temperature of -40°C , which promises the potential application of Zintl Li-Si phase materials, particularly in portable hydrogen production devices or extremely outdoor environments (i.e. polar areas and plateau regions). Moreover, the high efficiency of Zintl-HORs reaction inspires the green anode recovery in addition to the intrinsic hydrogen production from the degraded Si-based LIBs through a developed "charge-hydrolysis-separation" technology, demonstrating superiorities in economy and sustainability to alternative technologies.

Being economic- and environmental-friendly as well as highly efficient, our proposed "charge-hydrolysis-separation" strategy is promising for the recycling of next-generation degraded LIB compared to traditional pyrometallurgy and

hydrometallurgy methods and the established anode recovery strategies. More details are as following:

1) The traditional pyrometallurgy and hydrometallurgy methods mainly focus on the recycling of Li and cathode materials, while the anode material becomes CO₂ footprint or is discarded as waste.

2) Current established anode recovery strategies involve cumbersome processes like discharging, mechanical-physical treatment, screening, electrode separation (*Journal of Hazardous Materials*, 2022, 439, 129678), etc. Crucially, separating the electrode often necessitates time-consuming leaching or pyrolysis to remove the binders between active materials and current collectors. The additional chemical reagents needed in leaching or the high energy consumption and hazardous byproducts in pyrolysis (e.g., HF, dioxins and fluorinated organic compounds) further escalate the cost of anode recovery and pose possible risks of environmental pollution. Moreover, these strategies struggle to effectively separate silicon and graphite components from spent next-generation Si/C LIBs.

3) For the specific work you mentioned (*Journal of Alloys and Compounds*, 2023, 947, 169548), a simplified model based on Li_xSn alloys-H₂O reactions was established to simulate the recycling of pure Sn anode. However, similar to the Si anode, practical Sn anode systems are invariably integrated with carbonaceous additives (e.g., highly conductive graphite) to alleviate the huge volume expansion and severe stress concentration during lithiation. While this composite architecture is essential for electrochemical stability, it poses a significant challenge for recycling because the hydration treatment of lithiated electrodes fails to selectively isolate metallic Sn from the carbon matrix. Specifically, the Sn species remains chemically inert under aqueous conditions and cannot be leached, whereas the carbon component also persists as an insoluble residue. This co-retention diminishes the recovery efficiency of both valuable materials of Sn and carbon.

4) In comparison, our proposed recovery method can achieve efficient separation between active materials and collector, without requiring chemical leaching or high-temperature pyrolysis. Additionally, it enables efficient separation of silicon and graphite, which can yield high purity graphite for the anode regeneration or other functional applications. Therefore, our proposed novel battery recycling technology based on the hydrolysis of Zintl Si-based alloys, improves the applicability of Si-based hydrogen production method and promises a strategy for guiding scrapped Si as a valuable resource to support a post-circular economy.

To enrich the overall context of the research, the related revision and literature have been integrated and cited into the revised manuscript. Please check the lines 46-47, 93-94 and 361-371, pages 3-4 and 15-16 with red highlights.

1. Figure 1(b), as only a qualitative illustration, doesn't add much to the article. It might be more appropriate as part of a TOC figure. The same is true of Figure 3(i).

Response: Thanks for your important comment. In sight to your suggestion, we have removed Figures 1(b) and 3(i) and further consolidated Figure 1(a) to the original

Figure 3 for better presentation of the research results. Please check Figure 2, page 10 of the revised manuscript.

2. A very important control experiment, which was not reported, would be to test hydrogen generation by Si nanoparticles with addition of lithium hydroxide or sodium hydroxide equivalent to the lithium or sodium content in the Zintl phase. This would rule out the possibility that lithium hydroxide is simply catalyzing reaction of Si.

Response: Thanks for your insightful and constructive suggestions. We have designed the control experiments to comprehensively evaluate the effect of alkali metals' products (including the LiOH, NaOH, CH₃OLi and CH₃ONa) on the hydrogen conversion of Si. In specific employing the synthesized Zintl Li₂₁Si₅ and NaSi alloys as references based on the principle of alkali metal atomic equivalence, we have tested the hydrogen generation performance of Si nanoparticles in both aqueous (1.494 M LiOH/0.315 M NaOH) and methanolic (1.494 M CH₃OLi/0.315 M CH₃ONa) alkaline media. The hydrogen production tests were carried out with 0.1 g Si nanoparticles and 10 ml reaction solution, while the reaction temperature was 25°C. As shown in Fig. S19a, Si nanoparticles exhibit the induction durations of 20 min and 15 min in the LiOH and NaOH solutions, respectively, attributing to the dissolution process of surface oxide layer. Afterwards, the Si nanoparticles experience extremely slow hydrogen generation kinetics. It delivers 1.149 L g⁻¹ H₂ and 1.192 L g⁻¹ H₂ within the ultralong reaction durations of 235 minutes and 145 minutes, respectively, while the corresponding maximal Si utilization rates are 65.8% and 68.3% (Fig.S19b). Meanwhile, the CH₃OLi and CH₃ONa solutions show no promoting effect on the hydrogen generation of Si. The huge difference of hydrogen evolution kinetics between Si-alkalis solutions systems and Zintl Li₂₁Si₅/NaSi alloys-H₂O/CH₃OH systems suggests that the ultra-fast and ultra-high Si-H₂ conversions of Zintl phases are dominantly attributed to the unpaired electron structure rather than the simple catalyzing of symbiotic alkali metals' products. This result is further confirmed by the hydrogen production tests of the designed H-Li₂₁Si₅ composite (Figs. S20-22).

Fig. S19 (a) Hydrogen evolution curves of Si nanoparticles in 1.494 M LiOH solution, 0.315 M NaOH solution, 1.494 M CH₃OLi solution and 0.315 M CH₃ONa solution and (b) the corresponding maximal Si utilization rates.

The related revision has been consolidated in lines 263-265, page 11 of the revised manuscript and Fig. S19, Note S8, page 22 of the revised Supplementary information. Please check the red highlights.

3. What are the exothermicities of the various reactions and how much heating is observed? The authors should at least mention this, as managing the exothermicity would be critical at larger scale.

Response: Thanks for your constructive comment. We agree with the reviewer that exothermicity is very important for this reaction at the larger scale. Accordingly, we have calculated theoretical heating values and measured real-time temperature change of reaction systems, which have been added in the revised manuscript and Supplementary information.

Firstly, based on the hydrogen evolution performance of various Zintl Li-Si alloys in pure water and methanol at 25°C (Fig. 2a-d), the corresponding reaction equations were summarized in Table S6 and the theoretical exothermicity of these reactions was calculated. As shown in Fig. S12, 1g alloys of $\text{Li}_{21}\text{Si}_5$, $\text{Li}_{13}\text{Si}_4$, Li_7Si_3 and $\text{Li}_{12}\text{Si}_7$ in pure water and methanol can produce 16.1 kJ/ 16.9 kJ, 14.7 kJ/ 14.9 kJ, 13.4 kJ/ 10.2 kJ, 11.4 kJ/ 8.5 kJ of heat, respectively, highlighting the vigorously exothermic characteristic of these hydrogen production systems.

Furthermore, we employed an infrared thermal camera to monitor practical temperature changes in both liquid reaction area (L area) and flask area (F area) of different hydrolysis and methanolysis reactions over a 5-minute timeframe, under the conditions of 10 ml solvent and 0.1 g alloy in a 20 ml flask. As shown in Fig. S13a, the $\text{Li}_{21}\text{Si}_5$, $\text{Li}_{13}\text{Si}_4$, Li_7Si_3 and $\text{Li}_{12}\text{Si}_7$ alloys in pure water achieve peak L area temperatures of 50.4°C, 48.3°C, 47.5°C and 43.4°C, respectively, while corresponding flask temperatures don't exceed 46°C (Fig. S13b). Similarly, methanol-mediated reactions also produce substantial heat release (Fig. S13c, d). Notably, the $\text{Li}_{21}\text{Si}_5\text{-CH}_3\text{OH}$ system registers the most pronounced thermal output, while the maximum temperatures of L area and F area are 51.1°C and 52.3°C, respectively. Moreover, the maximum L area temperatures and relevant flask temperatures of $\text{Li}_{13}\text{Si}_4$, Li_7Si_3 and $\text{Li}_{12}\text{Si}_7$ alloys in methanol are 49.9°C/50.9°C, 48.7°C/47.1°C and 39.7°C/35.1°C, respectively.

Fig.S12 Theoretical heating values released from different Zintl Li-Si alloys in pure water and methanol.

Fig. S13 Temperature change curves of liquid (L) reaction area for various alloys and the IR images of the liquid areas and flasks with recorded maximum temperature: (a, b) hydrolysis systems and (c, d) methanolysis systems.

Table S6 The theoretical exothermicity of different Li-Si alloys in pure water and methanol.

Reaction equation	ΔH (kJ mol ⁻¹)	Heating (kJ g _{alloy} ⁻¹)
$\text{Li}_{12}\text{Si}_7 + 21.24\text{H}_2\text{O} \rightarrow 12\text{LiOH} + 4.62\text{SiO}_2 + 2.38\text{Si} + 15.24\text{H}_2$	-3190.7	11.4
$\text{Li}_7\text{Si}_3 + 11.848\text{H}_2\text{O} \rightarrow 7\text{LiOH} + 2.424\text{SiO}_2 + 0.576\text{Si} + 8.348\text{H}_2$	-1777.5	13.4
$\text{Li}_{13}\text{Si}_4 + 19.688\text{H}_2\text{O} \rightarrow 13\text{LiOH} + 3.344\text{SiO}_2 + 0.656\text{Si} + 13.188\text{H}_2$	-2986.4	14.7
$\text{Li}_{21}\text{Si}_5 + 29.69\text{H}_2\text{O} \rightarrow 21\text{LiOH} + 4.345\text{SiO}_2 + 0.655\text{Si} + 19.19\text{H}_2$	-4600.0	16.1
$\text{Li}_{12}\text{Si}_7 + 19.392\text{CH}_3\text{OH} \rightarrow 12\text{LiOCH}_3 + 1.848\text{Si}(\text{OCH}_3)_4 + 5.152\text{Si} + 9.696\text{H}_2$	-2373.4	8.5
$\text{Li}_7\text{Si}_3 + 10.72\text{CH}_3\text{OH} \rightarrow 7\text{LiOCH}_3 + 0.93\text{Si}(\text{OCH}_3)_4 + 2.07\text{Si} + 5.36\text{H}_2$	-1349.5	10.2
$\text{Li}_{13}\text{Si}_4 + 26.712\text{CH}_3\text{OH} \rightarrow 13\text{LiOCH}_3 + 3.428\text{Si}(\text{OCH}_3)_4 + 0.572\text{Si} + 13.356\text{H}_2$	-3020.7	14.9
$\text{Li}_{21}\text{Si}_5 + 40.62\text{CH}_3\text{OH} \rightarrow 21\text{LiOCH}_3 + 4.905\text{Si}(\text{OCH}_3)_4 + 0.095\text{Si} + 20.31\text{H}_2$	-4830.8	16.9

^a $\text{Li}_{21}\text{Si}_5$, ΔH : -555.2 kJ mol⁻¹ | Factstage software for |;

^b $\text{Li}_{13}\text{Si}_4$, ΔH : -417.7 kJ mol⁻¹ | Factstage software for |;

^c Li_7Si_3 , ΔH : -258.2 kJ mol⁻¹ | Factstage software for |;

^d $\text{Li}_{12}\text{Si}_7$, ΔH : -450.1 kJ mol⁻¹ | Factstage software for |;

^e H_2O , ΔH : -285.8 kJ mol⁻¹ | Factstage software for |;

^f CH₃OH, ΔH : -238.7 kJ mol⁻¹ | Factstage software for |;
^h LiOH, ΔH : -464 kJ mol⁻¹ | Factstage software for |;
ⁱ LiOCH₃, ΔH : -433 kJ mol⁻¹ | Ref. (*Journal of Organometallic Chemistry*, 1993, 460, 131-138) for |;
^j SiO₂, ΔH : -896.8 kJ mol⁻¹ | Factstage software for |;
^k Si(OCH₃)₄, ΔH : -1221 kJ mol⁻¹ | Ref. (*Journal of Organometallic Chemistry*, 1988, 345, 27-38) for |;
^l Si, ΔH : 0 kJ mol⁻¹;
^m H₂, ΔH : 0 kJ mol⁻¹;

The related revision has been consolidated in lines 199-201, page 9 of the revised manuscript and Figs. S12-13, Notes S2-3, Table S6, pages 14-15, 45 of the revised Supplementary information. Please check the red highlights.

4. The authors use absorbance at 645-650 wavenumbers as evidence of Si-H bonds, but we would usually identify Si-H based on stretching modes near 2200 wavenumbers. Were those peaks evident? Or is that spectral range inaccessible for some reason?

Response: Thanks for your constructive comment. We agree with the reviewer that the stretching peak at ~2200 cm⁻¹ is the characteristic peak for identifying the Si-H structure in many reports. However, this peak in our FT-IR spectra is rather weak mainly due to the solvent interference effect.

In specific, the FTIR peaks located at 645-650 cm⁻¹ and near 2200 cm⁻¹ are attributed to the bending mode and stretching mode of Si-H structure, respectively. Generally, the stretching peak at ~2200 cm⁻¹ is the characteristic peak for identifying the Si-H structure. However, our *in situ* FTIR measurements occurred at the solid-liquid bi-phase interface. The presence of a significant amount of water or CH₃OH solvent molecules can induce strong solvent absorption interference at the high wavenumber area, which increases the background intensity and peak width of the Si-H stretching mode, leading to a substantial decrease in its intensity ^[1]. Moreover, the hydrogen bonding structure between Si-H bond and solvents could also result to the reduction of vibrational frequency of the Si-H stretching mode ^[2]. In contrast, the peak intensity of the Si-H bending mode is less affected by solvent molecules owing to the low absorption interference of solvent molecules at the fingerprint region. Consequently, only a very weak Si-H stretching peak at around 2090 cm⁻¹ is observed in the methanolysis system (Fig. R1), while it is absent in the hydrolysis system due to the stronger interference from water compared to methanol. This phenomenon is consistent with previous reports ^{3,4}, where the intensity of the Si-H stretching mode near 2090 cm⁻¹ is significantly weaker than that of the bending mode due to the presence of solvent molecules. Furthermore, we designed a control experiment to confirm this result, taking the HSi(OCH₃)₃ as the model. As shown in Fig. R2, while pure HSi(OCH₃)₃ exhibits a strong Si-H stretching peak at ~2200 cm⁻¹, the CH₃OH/ HSi(OCH₃)₃ mixture shows negligible evidence of this peak. In contrast, the Si-H bending peak at 645-650 cm⁻¹ is clearly observable in both systems.

Fig. R1 FTIR spectra of $\text{Li}_{21}\text{Si}_5\text{-CH}_3\text{OH}$ system at different stages of *in situ* FTIR measurement.

Fig. R2 FTIR spectra of CH_3OH , $\text{HSi}(\text{OCH}_3)_3$ and $\text{CH}_3\text{OH}/\text{HSi}(\text{OCH}_3)_3$ mixture.

References

- [1] Y.C. Chen, H.S. Wang, J.Z. Umemura. A New Method to Obtain Fourier Transform Infrared Spectra Free from Water Vapor Disturbance. *Applied Spectroscopy*, 2010, 64, 1186-1189.
- [2] B. Stuart. *Infrared Spectroscopy: Fundamentals and Applications*. New York: John Wiley & Sons, Ltd. US, 2004
- [3] A.C. Dillon, P. Gupta, M.B. Robinson, A.S. Bracker and S.M. George. *Journal of Electron Spectroscopy and Related Phenomena*. FTIR Studies of Water and Ammonia Decomposition on Silicon Surfaces, 1990, 54-55, 1085-1095.
- [4] P. Gupta, A.C. Dillon, A.S. Bracker and S.M. George. FTIR studies of H_2O and D_2O decomposition on porous silicon surfaces. *Surface Science*, 1991, 245, 360-372.

5. The authors should more clearly present the fraction of hydrogen generation attributable to Li and to Si in the main manuscript. I was somewhat confused by Figure S15(c). The authors should clarify what they mean by “total” in that figure or label the two parts of each bar as Li and Si. For practical purposes, comparisons on a mass basis are more relevant than on a molar basis (and a mass basis is used elsewhere).

Response: Thanks for your constructive comment. In the revised Supplementary information, we clearly presented the fractions of hydrogen generation attributable to Li and to Si and updated Figure S15(c) and Table S3 accordingly.

In sight of your advice, we summarized a table for presenting the fraction of hydrogen generation attributable to Li and to Si, as shown in Table S3. Moreover, to avoid the confusion about the original Fig. S15(c), we used the mass basis for comparison rather than the molar basis and simplified the comparison forms. As shown in Fig. S21c, the theoretical hydrogen contributions of LiH and Si in the H-Li₂₁Si₅ are 1.674 L g⁻¹ and 0.798 L g⁻¹, while the total hydrogen yields of H-Li₂₁Si₅ in pure water and methanol are 1.988 L g⁻¹ and 1.674 L g⁻¹, respectively. Correspondingly, the Si utilization rates of H-Li₂₁Si₅ in water and methanol are 39.3% and 0%, respectively, markedly lower than those of Zintl Li₂₁Si₅ alloy.

Fig. S21c Theoretical hydrogen contribution of LiH/Si in H-Li₂₁Si₅ and the hydrogen yields of H-Li₂₁Si₅ in pure water/ methanol.

Table S3 Related hydrogen performance information used for the calculation of maximal Si utilization rates.

Temperature (°C)	Reaction systems	Maximal H ₂ Yield (L g ⁻¹)	H ₂ yield of Li species (L g ⁻¹)	H ₂ yield of Si species (L g ⁻¹)	Theoretical H ₂ yield of Si species (L g ⁻¹)
25	Li ₂₁ Si ₅ -H ₂ O	1.643	0.899	0.744	0.856
	Li ₁₃ Si ₄ -H ₂ O	1.595	0.786	0.809	0.968
	Li ₇ Si ₃ -H ₂ O	1.539	0.645	0.894	1.107
	Li ₁₂ Si ₇ -H ₂ O	1.334	0.525	0.809	1.225
	Li ₂₁ Si ₅ -CH ₃ OH	1.739	0.899	0.840	0.856
	Li ₁₃ Si ₄ -CH ₃ OH	1.616	0.786	0.830	0.968
	Li ₇ Si ₃ -CH ₃ OH	0.988	0.645	0.343	1.107

	Li ₁₂ Si ₇ - CH ₃ OH	0.849	0.525	0.324	1.225
	NaSi-H ₂ O	0.911	0.224	0.687	1.011
	NaSi-CH ₃ OH	0.866	0.224	0.642	1.011
10		1.754		0.855	
20	Li ₂₁ Si ₅ -	1.747		0.848	
30	CH ₃ OH	1.691	0.899	0.792	0.856
40		1.656		0.757	
10		1.566		0.921	
20	Li ₇ Si ₃ -H ₂ O	1.553		0.908	
30		1.521	0.645	0.876	1.107
40		1.396		0.751	

Thanks again for your insightful suggestions that effectively improve the quality of our manuscript. The related revision has been consolidated in Fig. S21c, Note S10, page 24 and Table S3, page 40 of the revised Supplementary information. Please check the red highlights.

6. The authors should further explain the negative activation barriers that they report – how are these “barriers” if they are negative. Is the structure a local maximum on the reaction path? If so, there should be a local minimum between the reactions and this nominal transition state structure. Adding the local minimum to the energy diagrams (Fig. 5e-f) would be helpful.

Response: Thanks for your constructive comment. The barriers are negative owing to the exceptionally intense electron interactions between solvent molecules and Zintl Li-Si alloy. We confirmed that the reaction barrier structure is the local maximum in the reaction path. The local minimum between the reactions and this nominal transition state structure is found and we have added the minimum to the energy diagrams in the revised manuscript.

In specific, the negative activation barrier indicates exceptionally intense electron interactions between solvent molecules and Zintl Li-Si alloy, resulting in the spontaneous breakage of H-O bond of H₂O or CH₃OH molecules during the reaction process. Meanwhile, we further carefully retrieved all calculation process and confirmed that the reaction barrier structure is the local maximum in the reaction path. Moreover, we also found that there is a local minimum between the reactions and transition state on the reaction path, which attributes to the rearrangement of H₂O/CH₃OH molecules on the surface of reactions (Figs. 4e, f), similar to the discussion of Tachibana’s work (*Journal of the American Chemical Society*, 1987, 109, 1383-1387).

Fig. 4 Energy profiles of (e) H₂O and (f) CH₃OH dissociation process on Si and various Zintl Li-Si alloy surfaces.

The related revision has been consolidated in lines 297-299, 313-315, Figs. 4e, f, pages 13-14 of the revised manuscript and Figs. 25-32, pages 28-35 of the revised Supplementary information. Please check the red highlights.

7. I noticed that the results of refs. 18 and 19 of the main manuscript were not included in the comparison table in the SI (table S4). The authors should ensure they have included comparisons to all prior reports if they want to claim to have outperformed all prior methods. Ref. 18 focuses mainly on the speed of hydrogen generation but also claims to produce more than 2.0 moles of dihydrogen per mole of silicon, attributing the excess to the hydrogenated surface of nanoparticles. This would imply 100% silicon utilization. Similarly, ref. 19 focuses on a potential biomedical application, but seems to claim 100% Si utilization, with > 2 moles dihydrogen per mol Si, again attributable to hydrogenated surfaces.

Response: Thanks for your constructive comment. We have included refs. 18 and 19 in the revised Table S4 and Fig. S9 for the overview comparison with our work in terms of material synthesis efficiency, reaction solution safety, total hydrogen yield, average hydrogen production rate and Si atom utilization rate. While we recognize the good hydrogen generation speed and 100% silicon utilization in refs. 18 and 19 due to the hydrogenated surface structure of nanoparticles and the use of corrosive solution (8 M KOH solution for ref. 18 and ~pH = 12 NaOH solution for ref. 19), our designed Li₂₁Si₅ based hydrogen systems are superior in terms of preparation feasibility with high efficiency, safety and environmental-friendliness, as well as wide temperature range covering ultra-low temperature at -40°C, which are especially promising for sustainable hydrogen conversion with high performance-to-cost ratio.

Thanks again for your kind suggestion, the related works of refs. 18 and 19 have been supplemented into Fig. S9, Table S4, pages 11, 41-42 of revised Supplementary Materials. Please check the red highlights.

8. The methods section is inadequate. For example, “mixture of Li sheet and Si powder with Li/Si ratio of 21:5 was heated and continuously stirred to become well-distributed.” Heated to what temperature? Stirred how? In what form were the FTIR measurements made? The authors should provide a level of detail sufficient for someone to reproduce their work.

Response: Thanks for your valuable comment. We have supplemented the experimental details in the methods section of the revised manuscript.

In specific, the heating temperature was 250°C; the stirring process was proceeded until the Li metal well distributed on the surface of Si powder, ultimately resulting in a black product; All FTIR measurements were carried out with an attenuated total reflection model (ATR-FTIR). Moreover, additional experimental details have been incorporated into the Methods section to enhance the reproducibility of this work.

The related revision has been consolidated in the Methods section, pages 17-19 of the revised manuscript, please check the red highlights.

9. The authors should clearly indicate in the results section the quantity of excess water or methanol being used (e.g., a 100:1 mass ratio of water to solids). Does the process require this much excess water? Did they do any experiments at lesser excess water?

Response: Thanks for your valuable comment. We have clarified the amount of solvent used in the “result section” of the revised manuscript. To be short, the current water volume optimizes the performance in terms of reaction dynamics and ultimate hydrogen yield. We also performed control experiments at the lesser excess water, as shown in Fig. S14.

According to the theoretical reaction equation of $\text{Li}_{21}\text{Si}_5 + 31\text{H}_2\text{O} \rightarrow 21\text{LiOH} + 5\text{SiO}_2 + 20.5\text{H}_2$, various stoichiometric amounts of pure water ($\text{H}_2\text{O}/\text{Li}_{21}\text{Si}_5 = 62\sim 248$ mol/mol, equivalent to 0.39~1.56 mL $\text{H}_2\text{O}/0.1$ g $\text{Li}_{21}\text{Si}_5$) were added to react with $\text{Li}_{21}\text{Si}_5$ alloy at 25°C. Under these conditions of lesser excess water, as revealed in Fig.S14, the $\text{Li}_{21}\text{Si}_5$ alloy demonstrates steep hydrogen generation kinetics during the initial 10-second process and delivers 1.532 L g^{-1} , 1.578 L g^{-1} and 1.446 L g^{-1} H_2 within 1 min in the 0.39 mL, 0.78 mL and 1.56 mL water, respectively. These hydrogen evolution behaviors show significant deviation from the performance observed in the much excess water condition (~10 ml water), particularly regarding reaction dynamics and ultimate hydrogen yield. The critical difference stems from water's bifunctional nature as both reactant and thermal buffer, while its limited quantity dramatically affects heat dissipation capacity during the exothermic hydrolysis process. The intensified thermal accumulation not only initiates a self-accelerating effect on the initial hydrogen release, but also produces uncontrolled local temperature escalation that induces partial melting of the alloy matrix and facilitates rapid precipitation of passivation products, further blocking water penetration and terminating the hydrolysis reaction prematurely. Especially, even the initial reactivity is slightly reduced when the water volume increases to 1.56 mL, the thermal runaway and passivation layer formation still fail to be completely suppressed (as seen in the inset image of Fig.S14), conversely resulting in a lower hydrogen yield. Comparatively, the water-excessive

system (10 mL) demonstrates superior thermal regulation and maintains moderate temperature spikes. This thermal stabilization enables a higher reaction efficiency, yielding $1.643 \text{ L g}^{-1} \text{ H}_2$ through sustained hydrolysis within 60 seconds. The delicate balance between thermal management and reaction progression highlights the crucial role of the amount of solvent in controlling both the temperature of reaction system and the hydrogen evolution kinetics.

Fig. S14 The hydrogen production curves of $\text{Li}_{21}\text{Si}_5$ alloy with various amounts of pure water at 25°C , the inset image is the product of the reaction system with 1.56 mL pure water.

The related revision has been consolidated in lines 201-204, page 9 of the revised manuscript and Fig. S14, Note S4, pages 16-17 of the revised Supplementary information. Please check the red highlights.

10. The authors should address the air stability (particularly with significant humidity) and hazards of the lithium-silicon materials. They may want to consult an SDS for a commercial lithium-silicon powder used in batteries ([https://www.eaglepicher.com/sites/default/files/EHS-SDS-1005 Lithium Silicon powder SDS rev Orig 5-10-18.pdf](https://www.eaglepicher.com/sites/default/files/EHS-SDS-1005_Lithium_Silicon_powder_SDS_rev_Orig_5-10-18.pdf)).

Response: We appreciate the reminder of the reviewer for the air stability and hazard of lithium-silicon materials. As you are concerned, the Li-Si alloy is prone to failure in humid air and may even burn. Accordingly, we have designed a paraffin coating method to improve the air stability and operation safety of $\text{Li}_{21}\text{Si}_5$ in the revised manuscript, while the super-hydrophobic paraffin coating can effectively mitigate moisture-induced degradation of the internal alloy by acting as a barrier against humid air.

Paraffin coated $\text{Li}_{21}\text{Si}_5$ composite ($\text{Li}_{21}\text{Si}_5\text{@PA}$) preparation: 0.005g PA was dissolved in 1 mL n-heptane and the mixture was then applied dropwise to cover the entire surface of a 0.1 g $\text{Li}_{21}\text{Si}_5$ tablet. The PA can spontaneously precipitate upon the rapid evaporation of n-heptane solvent, thus forming a uniform PA coating film on the surface of alloy tablet within 10 s.

Detailed characterizations confirm the composite structure and coating quality. XRD pattern of $\text{Li}_{21}\text{Si}_5@PA$ shows the peaks corresponding to $\text{Li}_{21}\text{Si}_5$ and PA, indicating decent compatibility between the active $\text{Li}_{21}\text{Si}_5$ alloy and PA coating (Fig. S15a). EDS mapping and TOF-SIMS results reveal the uniformity and distribution of the PA coating layer. As shown in Fig.S15b, surface EDS mapping demonstrates an even distribution of C and Si elements for the $\text{Li}_{21}\text{Si}_5@PA$. Moreover, TOF-SIMS depth profiling and three-dimensional images suggest that the dense PA coating layer is dominantly located on the outer surface of the alloy matrix (Figs. S15c, d). As a result, the well-defined coating structure would promote $\text{Li}_{21}\text{Si}_5@PA$ to achieve splendid air stability.

The air exposure tests of $\text{Li}_{21}\text{Si}_5$ alloy and $\text{Li}_{21}\text{Si}_5@PA$ composite were conducted under ambient atmosphere (64-70% humidity, $\sim 28^\circ\text{C}$). As shown in Fig.S16a, severe structural degradation of $\text{Li}_{21}\text{Si}_5$ alloy is evident just after 10 s air exposure. When the exposure time is prolonged to 1h, the active $\text{Li}_{21}\text{Si}_5$ phase structure is almost poisoned, accompanied by the significant formation of LiOH and Li_2CO_3 (Fig. S16b). In contrast, the $\text{Li}_{21}\text{Si}_5@PA$ shows a much more stable structure, exhibiting only slight surface failure after the same 1-hour exposure period (Figs. S16a, b). Therefore, while the initial hydrogen production kinetics of $\text{Li}_{21}\text{Si}_5@PA$ is slightly lower than the $\text{Li}_{21}\text{Si}_5$ alloy ($1.552 \text{ L g}^{-1} \text{ H}_2$ in 2 min vs $1.643 \text{ L g}^{-1} \text{ H}_2$ in 1 min), it achieves a significant enhancement of hydrogen yield retention from 58.4% to 96.6% (Fig. S16c).

Fig. S15 (a) XRD pattern of $\text{Li}_{21}\text{Si}_5@PA$ composite. (b) EDS mapping of $\text{Li}_{21}\text{Si}_5@PA$ composite. (c) The vertical distribution profiles of ^{12}C , ^7Li , and ^{28}Si signals and (d) corresponding three-dimensional mapping images of $\text{Li}_{21}\text{Si}_5@PA$ composite.

Fig. S16 (a) Digital photos of Li₂₁Si₅ alloy and Li₂₁Si₅@PA composite after air exposure within 10 s and 1 h; (b) XRD patterns of Li₂₁Si₅ alloy and Li₂₁Si₅@PA composite after air exposure within 1 h; (c) Hydrogen evolution curves of pristine Li₂₁Si₅@PA composite, Li₂₁Si₅ alloy and Li₂₁Si₅@PA composite after air exposure within 1 h.

The related revision has been consolidated in lines 225-231, pages 9-10 of the revised manuscript and Figs. S15-16, Notes 5-6, pages 18-19 of the revised Supplementary information. Please check the red highlights.

Reviewer #2 (Remarks to the Author):

This work reports the Zintl phase Si-based alloys featuring discrete Si clusters with substantial unpaired electrons, demonstrating superior hydrogen production yields and enhanced Si utilization efficiency in water/methanol systems across a broad operational temperature range. The prominent unpaired electron configuration in these alloys achieves exceptional interaction with solvent molecules, thereby remarkably reducing the activation barriers of H₂O/CH₃OH dissociation as evidenced by DFT calculation and in situ FTIR tests, which primarily accounts for the hydrogen production performance and Si utilization rate surpassing those reported in previous studies. This methodology carries significant implications for designing high-performance Si-based hydrogen production systems, particularly given its straightforward material preparation and environmentally benign reaction conditions. Furthermore, it provides novel insights into the sustainable recovery of valuable resources from spent LIBs. This work is well-presented and thoroughly analyzed, which is a very interesting and high-quality study from which the portable hydrogen supply technology will benefit. I recommend its publication after addressing the following minor revisions.

Response: We appreciate the reviewer for the approval of the novelty and significance of our work. We also thank the reviewer for the valuable comments and suggestions. Here, we have carried out more detailed discussions to address all reviewer's concerns and improved the quality of our manuscript.

1. In Figure 3a and c, the hydrogen production process of Li₁₃Si₄, Li₇Si₃ and Li₁₂Si₇ alloys in water exhibit slow hydrogen kinetics after the initial rapid hydrogen evolution, however, this phenomenon is less pronounced in methanolysis systems. Could the authors elaborate on the potential origins of this solvent-dependent kinetic behavior?

Response: Thanks for your valuable comment. The observed kinetic disparity is primarily attributed to the different catalytic effects of the concomitant alkali metals' products. Firstly, whatever for the hydrolysis systems or methanolysis systems, the unpaired electron structures of the Zintl Li-Si alloys dominated the hydrogen production process and offer rapid initial hydrogen evolution. However, as for the hydrolysis process, the concomitant LiOH solution can catalyze the reaction between residual silicon and water, sustaining slow hydrogen generation until complete passivation of the silicon species occurs. As for the methanolysis, the concomitant CH₃OLi shows no catalytic activity toward the silicon-methanol reaction, as also demonstrated in the Figs. S19-22, therefore showing no further hydrogen production process.

To enrich the overall context of the research, the related revision has been integrated into the revised manuscript, please check the lines 167-176, page 8 within red highlights.

2. In Fig.5g, the ΔG_{MA} shows a linear correlation with the number of the charge transferred in the Li-Si alloys. The authors should offer additional physical insights to give the readers a better understanding.

Response: Thanks for your valuable suggestion. We have added the discussion on the physical mechanism of the linear correlation of ΔG_{MA} in the revised manuscript.

Based on the DFT calculations, it can be observed that the activation barrier energy reductions (ΔG_{MA}) show strong linear correlations with the number of electrons transferred between Li and Si, irrespective of whether the reaction occurs in H₂O or CH₃OH. This trend further underscores that the intrinsic electronic transfer effect between Li and Si dominates the activation kinetics of solvent decomposition. Moreover, the robust linear ΔG_{MA} -electron transfer correlations provide a quantifiable descriptor for rational alloy design and reaction kinetics adjustment based on various practical hydrogen generation demands.

To enrich the overall context of the research, the related revision has been integrated into the revised manuscript, please check the lines 305-309, page 13 within red highlights.

3. While Li-Si and NaSi alloys were employed as models to investigate the role of unpaired electron structures in Si-H₂ conversion, could this strategy be rationally extended to other Si-based alloy systems?

Response: Thanks for your constructive comment. In fact, this strategy should also be available for other Zintl alkalis-Si alloys with similar unpaired electron structures and bonding forms, such as K₁₂Si₁₇, Rb₁₂Si₁₇, Rb₇NaSi₈ and Cs₁₂Si₇ alloys. As demonstrated by the report of Nolan's et al. (*Inorganic Chemistry*, 2015, 54, 396-401), these alloys can react with the reagents containing positive H* (such as NH₄Br) to form active Si-H intermediate containing negative H*. This aligns well with our work, where the positive H* of hydroxide radical in H₂O/CH₃OH molecules is induced to form active

Si-H intermediate through the electron interaction mechanism. Therefore, these findings indicate that similar Zintl alkali-Si alloys hold promise for achieving efficient Si-H₂ conversion via reacting with H₂O/CH₃OH molecules.

4. Abbreviations such as "TMOS" (line 259, page 11) must be explicitly defined upon their first occurrence. Please ensure consistent adherence to this convention throughout the manuscript.

Response: Thanks for your valuable comment. In the revised manuscript, we have carefully checked and ensured that all the abbreviations are defined upon their first occurrence. In specific, TMOS represents tetramethoxysilane.

5. The detailed information regarding the chemicals and materials used should be provided.

Response: Thanks for your valuable comment. We have supplemented a "Materials and Chemicals" part into the "Methods" section, which offers detailed information of the chemicals and materials used in this work.

The related revision has been consolidated in "Materials and chemicals" part, pages 17-18 of the revised manuscript, please check the red highlights.

6. In the Method section, the authors are encouraged to provide additional details regarding the sample preparation, such as the specific method for removing residual n-heptane following the ball milling process.

Response: Thanks for your valuable comment. The residual n-heptane was effectively removed by heating the milled products in the glove box at 80°C for 30 min. In addition, more experimental details have been supplemented in the Methods section of the revised manuscript to improve the reproducibility of our work.

The related revision has been consolidated in the Methods section, pages 18-19 of the revised manuscript, please check the red highlights.

Response to the reviewers' comments

Dear Reviewers,

We highly appreciate your constructive comments and the endorsement of our revised manuscript. We have carefully revised our manuscript according to your new suggestions and comments. The revised places were highlighted and the point-to-point replies are as follows.

Reviewer #1 (Remarks to the Author):

The authors have, for the most part, addressed my concerns with the original version of the paper. Their explanations in the response to the reviews are quite helpful, but in most cases the changes in the manuscript itself were minor. For example, the abstract was unchanged and still leads off with a statement about "cost-effective" hydrogen generation. I still believe that hydrogen generation will only be cost-effective in a few niche applications. The value proposition, if there is one, will be in the battery recycling application, where the value will come from recovery of the Si, Li, and carbon-based components and from reduced waste generation.

That being said, with the open peer review model of Nature Communications, I think the paper is acceptable for publication. A sufficiently-interested reader could dig into the peer review file for the detailed discussion provided in the response to reviewers.

Response: We sincerely thank the reviewer for their positive opinion and endorsement of our revised manuscript. We have rephrased the abstract and main text in terms of the cost-effective description for hydrogen generation.

Revised abstract: “The hydrolysis/methanolysis of silicon has received considerable attention to achieve efficient and on-demand hydrogen conversion.”

Revised text in page 7, line 163: “When considering the hydrogen yield alongside hydrolysis rate, the Zintl Li_7Si_3 alloy emerges as the superior option under pure water conditions.”

Reviewer #2 (Remarks to the Author):

All of the issues have been well addressed in the revised manuscript, and I think the manuscript can be accepted for publication now.

Response: We sincerely thank the reviewer for their positive opinion and endorsement of our revised manuscript.